# EGFR-Specific Tyrosine Kinase Inhibitor Modifies NK Cell-Mediated Antitumoral Activity against Ovarian Cancer Cells

**DOI:** 10.3390/ijms20194693

**Published:** 2019-09-22

**Authors:** Nina Mallmann-Gottschalk, Yvonne Sax, Rainer Kimmig, Stephan Lang, Sven Brandau

**Affiliations:** 1Department of Gynecology and Obstetrics, University of Duisburg-Essen, Hufelandstr. 55, 45147 Essen, Germany; rainer.kimmig@uk-essen.de; 2Department of Otorhinolaryngology, University of Duisburg-Essen, Hufelandstr. 55, 45147 Essen, Germany; Yvonne.Sax@uk-essen.de (Y.S.); stephan.lang@uk-essen.de (S.L.); sven.brandau@uk-essen.de (S.B.)

**Keywords:** NK cell, cetuximab, erlotinib, ovarian cancer, cancer immunotherapy, ADCC

## Abstract

The adverse prognosis of most patients with ovarian cancer is related to recurrent disease caused by resistance to chemotherapeutic and targeted therapeutics. Besides their direct activity against tumor cells, monoclonal antibodies and tyrosine kinase inhibitors (TKIs) also influence the antitumoral activity of immune cells, which has important implications for the design of immunotherapies. In this preclinical study, we treated different ovarian cancer cell lines with anti-epidermal growth factor receptor (EGFR) TKIs and co-incubated them with natural killer (NK) cells. We studied treatment-related structural and functional changes on tumor and immune cells in the presence of the anti-EGFR antibody cetuximab and investigated NK-mediated antitumoral activity. We show that long-term exposure of ovarian cancer cells to TKIs leads to reduced responsiveness of intrinsically sensitive cancer cells over time. Inversely, neither long-term treatment with TKIs nor cetuximab could overcome the intrinsic resistance of certain ovarian cancer cells to anti-EGFR agents. Remarkably, tumor cells pretreated with anti-EGFR TKIs showed increased sensitivity towards NK cell-mediated antibody-dependent cellular cytotoxicity (ADCC). In contrast, the cytokine secretion of NK cells was reduced by TKI sensitization. Our data suggest that sensitization of tumor cells by anti-EGFR TKIs differentially modulates interactions with NK cells. These data have important implications for the design of chemo-immuno combination therapies in this tumor entity.

## 1. Introduction

Ovarian cancer still remains a gynecological malignancy with an adverse prognosis [1]. Radical resection followed by platinum-taxane-based chemotherapy recently combined with the anti-VEGF-inhibitor bevacizumab has improved prognosis in the last decade [2]. Nevertheless, most patients with initially advanced stages suffer from relapse and display limited long-term survival. The high recurrence rate is due to the cumulative resistance to common chemotherapeutic regimens and the lack of effective innovative therapeutic concepts. Thus, targeted therapies and immunotherapeutic approaches are being intensively tested in clinical and preclinical studies. 

Among potential targets for the treatment of ovarian cancer, the epidermal growth factor receptor (EGFR) may be promising since EGFR is overexpressed in most ovarian cancers and is correlated to poor prognosis [3,4]. In in vitro and in vivo studies, the chimeric monoclonal antibody cetuximab displays antiproliferative and apoptotic effects and is able to reduce chemotherapy resistance in various human cancer entities including ovarian cancer [5,6]. In consequence, cetuximab has been approved for the treatment of metastatic colorectal cancer and advanced head and neck cancer [7,8]. However, in unselected ovarian cancer populations, cetuximab showed only minimal activity either as a single agent or in combination with chemotherapy [9,10,11]. Tyrosine kinase inhibitors (TKIs) are low molecular compounds which act intracellularly by binding to the tyrosine kinase domain and preventing intracellular signaling. Most clinical data are available for the reversible EGFR-specific TKIs erlotinib (Tarceva^®^) and gefitinib (Iressa^®^). In preclinical studies, erlotinib showed antiproliferative activity even in platinum-resistant ovarian cancer cell lines [6,12], and gefitinib displayed synergistic effects with cytostatic agents [13]. However, recent clinical trials failed to show a prognostic benefit for erlotinib or gefitinib in ovarian cancer either as single agents or in combination with chemotherapy or as maintenance therapy [14,15,16,17,18]. So far, anti-EGFR targeting in ovarian cancer has not reached sufficient clinical benefit.

Clinical efficacy may be limited due to intrinsic and extrinsic resistance to anti-EGFR targeted therapy. K-ras mutations and changed EGFR degradation and internalization may provoke resistance to anti-EGFR antibodies [19,20]. EGFRvIII is the best characterized mutant EGFR variant which seems to reduce the efficacy of anti-EGFR agents [21]. Due to the complementary mode of action, the dual blockade with anti-EGFR antibodies and anti-EGFR TKIs may abrogate resistance to single anti-EGFR agents. Clinical data of combined treatment in metastatic colorectal cancer are promising [22]. However, in lung cancer, acquired anti-EGFR TKI resistance could not be overcome by cetuximab [23]. Corresponding to this data, we showed in a preclinical study with different ovarian cancer cell lines that dual anti-EGFR blockade led to neither synergistic antiproliferative effect in anti-EGFR-sensitive cells nor de novo anti-EGFR sensitivity in anti-EGFR-resistant cell lines [24]. The significance of EGFR mutations remains unclear since there is evidence that, in non-small-cell lung cancer (NSCLC), mutations in the TK domain of EGFR are predictive for the response to gefitinib [25]. However, in ovarian cancer, specific TK-domain mutations seem to play an inferior role [6]. In conclusion, further mechanisms of anti-EGFR resistance as well as the relevance of dual anti-EGFR treatment in ovarian cancer remain to be elucidated.

Beyond their primary function of targeting tumor cells and mediating specific intracellular signaling effects, monoclonal antibodies may function as immune modifiers. They specifically bind to target cells and trigger antibody-dependent cellular cytotoxicity (ADCC) by binding the Fcγ-receptor on monocytes, granulocytes, and natural killer (NK) cells. This immune modifying role may even take place if functional resistance of the antibody’s specific target is predominant. 

NK cells can have substantial anti-tumor activity. During antibody-dependent or natural cytotoxicity, NK cells kill tumor cells by releasing perforin/granzymes or by activating apoptotic pathways [26]. Loss of MHC class I molecules or upregulation of stress-induced ligands such as MICA/B (MHC class I polypeptide-related sequence A/B) and UL16 binding proteins 1—6 (ULBP 1–-6) on target cells are crucial triggers inducing NK cell-mediated cytotoxicity. Thus, NKG2D and DNAM-1 as well as NKp46, NKp44, and NKp30 seem to be the central corresponding receptors on NK cells [27]. NK cells, via the secretion of various cytokines, support their antitumoral activity and regulate other adaptive immune cell responses (e.g., T-cell function). NK cell activity itself is adjusted by various regulatory receptors, such as killer-cell immunoglobulin-like receptors (KIRs) and killer cell lectin receptors (KLRs). Different cytokines like IL2, IL12, IL15, and IL18 as well as interactions with macrophages, dendritic cells (DCs), and mesenchymal stromal cells (MSCs) direct NK cell function [28,29,30,31]. Thus, NK cells may be a potent component in immunotherapeutic concepts.

Our own previous studies have revealed that ovarian cancer cells show predominant resistance to NK cell-mediated natural cytotoxicity [24,32]. Cetuximab was able to overcome natural NK cell resistance significantly, but the addition of EGFR-specific TKIs did not further augment this ADCC mechanism.

As specific TKIs are usually applied over the long term for oncological therapy, the present study aimed to examine the consequences of long-term treatment of anti-EGFR TKIs on ovarian cancer cells displaying intrinsic resistance to anti-EGFR antibodies. We examined whether ovarian cancer cells pretreated with anti-EGFR TKIs (“sensitization”) could regain susceptibility to anti-EGFR antibodies. Furthermore, we focused on NK cells and tested the effects of sensitization on different NK functions.

## 2. Results

### 2.1. Anti-EGFR Sensitization of Ovarian Cancer Cells Enhanced Tumor Cell Viability and Increased Resistance to Cetuximab

In previous studies, we demonstrated that most EGFR-positive cancer cell lines are resistant to the EGFR antibody cetuximab and also to related tyrosine kinase inhibitors (TKIs) [24]. Furthermore, we showed that the dual blockade with antibodies and TKIs could not overcome this resistance [24]. Therefore, in the present study we analyzed whether a long-term pretreatment of ovarian cancer cells with anti-EGFR TKIs for several days and weeks (“sensitization”) could enable or enhance susceptibility to the EGFR antibody cetuximab in terms of cell viability. For our studies, we selected the constitutively anti-EGFR-susceptible ovarian cancer cell line IGROV-1 and SKOV-3 with intrinsic resistance to cetuximab. For sensitization, we used the anti-EGFR TKIs erlotinib, gefitinib, and vandetanib.

First, cell viability was assessed under continued anti-EGFR TKI treatment after sensitization. Figure 1a shows the basic susceptibility of unsensitized IGROV-1 cells to erlotinib (E), gefitinib (G), and vandetanib (V). Addition of cetuximab did not reduce cell viability further (striped columns). Sensitization of IGROV-1 cells (sE, sG, sV) with anti-EGFR TKIs for 7 days (Figure 1b) and 6 weeks (Figure 1c) resulted in a significantly increased resistance of the cancer cells to TKI treatment over time (compare white columns in Figure 1a–c). Additional cetuximab (striped columns) was not able to overcome the developed resistance. 

In the next series of experiments, we tested the consequences of discontinuation of the TKI exposure after 7 days and 6 weeks on ovarian cancer cell viability. Grey columns in Figure 1b,c document a dramatic increase of cell proliferation of sensitized cancer cells, which was quantified 72 h after the completion of TKI treatment. Thus, the overwhelming cell proliferation was beyond the primary level of unsensitized tumor cells. However, under these conditions, added cetuximab could overcome resistance partially (grey striped columns). Nevertheless, comparing the TKI exposure for 7 days to 6 weeks in IGROV-1 cells, we observed that the decelerating influence of cetuximab decreased over time. 

In contrast, SKOV-3 cells showed an extensive resistance to single anti-EGFR TKI treatment as well as dual blockade with additional cetuximab (Figure 1d). Furthermore, the long-term anti-EGFR TKI sensitization for 7 days or 6 weeks was not able to overcome resistance and create susceptibility to cetuximab Figure 1e,f. 

### 2.2. Sensitization with Anti-EGFR TKI Decreased Sensitivity to FasLigand but Enhanced Ovarian Cancer Cells for NK Cell-Mediated Cytotoxic Degranulation 

Based on our present results of increasing resistance of anti-EGFR-sensitive ovarian cancer cells to cetuximab by anti-EGFR TKI sensitization, we further examined whether sensitivity of ovarian cancer cells to death receptor ligands was impaired by anti-EGFR TKI sensitization. Therefore, the rate of apoptosis of sensitized tumor cells was assessed after exposure to FasLigand and tumor necrosis factor-related apoptosis-inducing ligand (TRAIL). Indeed, we observed in erlotinib-sensitized IGROV-1 cells a significant increase of resistance to FasLigand in a dose-dependent manner (Figure 2a), whereas tumor cell sensitivity to TRAIL remained unaffected by sensitization (Figure 2b).

Besides the activation of death receptors, NK killing of tumor cells is mainly mediated via granzymes/perforin [26]. The following experiments concentrated on the impact of anti-EGFR agents on NK cell-mediated cytotoxic degranulation. In previous studies, we showed that most ovarian cancer cells displayed a distinct intrinsic resistance to natural NK cell cytotoxicity [24]. While the anti-EGFR antibody cetuximab succeeded in overcoming this NK resistance partially via ADCC (antibody-dependent cellular cytotoxicity), the short-term addition of anti-EGFR tyrosine kinase inhibitors (TKIs) failed to show any modulating effect [24]. Therefore, in the present study we co-incubated anti-EGFR sensitized ovarian cancer cells with NK cells and determined that CD107a expressed on NK cells during degranulation was a marker for granzyme/perforin-mediated cytotoxicity. As a parameter of NK cell-mediated tumor cell lysis, we used the flow cytometric measurement of 7-AAD-positive tumor cells.

Interestingly, anti-EGFR sensitization of IGORV-1 for 7 days resulted in significantly enhanced NK cytotoxicity (Figure 2c) and increased NK-specific tumor cell lysis (Figure 2d) in the presence of cetuximab. For SKOV-3, we also demonstrated significantly increased sensitivity to NK cytotoxicity (Figure 2e) and enhanced tumor cell lysis (Figure 2f) after 6 weeks of TKI treatment in the presence of cetuximab. In contrast, natural cytotoxicity in the absence of cetuximab remained unaffected by sensitization in both cell lines. 

### 2.3. Anti-EGFR TKI Sensitized Ovarian Cancer Cells Led to Reduced Cytokine Release of Secretory NK Cells 

Because we showed that anti-EGFR sensitized ovarian cancer cells enhanced NK cell degranulation, we further evaluated the effect of sensitized target cells on NK cytokine secretion activity. We determined the production and secretion of IFNγ of NK cells in the presence of cetuximab and sensitized ovarian cancer cells. 

As illustrated in Figure 3a, the detection of IFNγ-positive NK cells by flow cytometry in the presence of erlotinib-sensitized ovarian cancer cells and cetuximab was reduced significantly. Confirming this data, the frequency of IFNγ-positive NK cells in the ELISpot after co-incubation with sensitized ovarian cancer cells was also diminished significantly (Figure 3b). These data also show that the inhibitory effect of sensitized tumor cells on NK cytokine secretion was more pronounced in the absence of cetuximab.

### 2.4. Anti-EGFR Sensitized Ovarian Cancer Cells Show Altered Expression of Stress-Induced Ligands and MHC I and Enhanced Cytokine Release 

To further characterize the biological effects of anti-EGFR sensitization, we evaluated the expression of stress-induced ligands and MHC class I molecules as well as the release of different cytokines by sensitized ovarian cancer cells. 

As illustrated in Figure 4a–d, anti-EGFR sensitization resulted in a significant downregulation of MHC class I polypeptide-related sequence A (MICA) and MHC class I polypeptide-related sequence B (MICB) on both tested cell lines. In contrast, expression analyses of the NKG2D ligands UL16 binding proteins 1—6 (ULBP 1–-6) showed a variable non-significant downregulation on IGROV-1 and SKOV-3 cells.

As shown in Figure 4e,f, anti-EGFR sensitization led to downregulated MHC I in IGROV-1 cells and SKOV-3 cells, respectively. Additionally, protein analyses of the supernatants of sensitized tumor cells displayed a significant increase of the NK-activating cytokine IL18, as illustrated in Figure 5. 

In summary, anti-EGFR TKI sensitization of ovarian cancer cells led to enhanced functional resistance to anti-EGFR TKIs which could not be compensated by cetuximab sufficiently. Conversely, intrinsic resistance to cetuximab could not be resolved by sensitizing ovarian cancer cells with anti-EGFR TKIs. Essentially, anti-EGFR TKI sensitization of tumor cells caused decreased sensitivity to FasLigand but remarkably enhanced cytotoxic degranulation activity of co-incubated NK cells in the presence of cetuximab. Therefore, the downregulation of MHC I on tumor cells as well as enhanced release of the NK-stimulating cytokine IL18 out of ovarian cancer cells may be crucial. Of note, this enhanced susceptibility to NK cytotoxicity occurs despite downregulation of NKG2D ligands in sensitized ovarian cancer cells. However, secretory NK cells were functionally inhibited by sensitized ovarian cancer cells, especially in the absence of cetuximab. 

## 3. Discussion

Despite initial efficacy of the primary therapy, the adverse prognosis of advanced ovarian carcinoma is caused by the development of chemoresistance and high recurrence rate [1]. Targeted therapies with specific antibodies or tyrosine kinase inhibitors (TKIs) have been developed to overcome chemoresistance. Among these, the epidermal growth factor receptor (EGFR) may be a promising target. EGFR is overexpressed in most ovarian cancers, which is correlated with poor prognosis [3,4]. However, clinical studies evaluating specific anti-EGFR drugs in unselected ovarian cancer patient populations have failed so far to show a relevant clinical benefit [11,17]. In addition, our previous studies revealed that most ovarian cancer cell lines show functional resistance to the anti-EGFR antibody cetuximab which could not be resolved by the simultaneous addition of the corresponding anti-EGFR TKIs [24]. Thus, new approaches involving immunotherapeutic concepts in ovarian cancer should evaluate how the resistance of ovarian cancer cells to a specific therapy can be resolved.

In our own previous studies, we showed that the resistance of ovarian cancer cells to the anti-EGFR antibody cetuximab could be partially overcome by adding NK cells mediating antibody-dependent cellular cytotoxicity. However, the simultaneous addition of anti-EGFR TKIs could not augment this effect further [24]. Thus, in the present study we evaluated a long-term influence (“sensitization”) of anti-EGFR TKIs on ovarian cancer cells. First, we analyzed the biological effects of this sensitization on structural and functional changes of ovarian cancer cells. Then, we tested anti-EGFR-sensitive as well as anti-EGFR-resistant ovarian cancer cells. Second, we examined their sensitivity towards NK cells, which may represent potential partners in multimodal chemo-immuno combination therapies. 

Our data demonstrated that the sensitization of anti-EGFR-sensitive ovarian cancer cells with anti-EGFR TKIs caused increasing functional resistance. This development could not be sufficiently overcome by the addition of cetuximab. Furthermore, in anti-EGFR TKI-resistant ovarian cancer cells, anti-EGFR sensitization did not lead to de novo susceptibility to anti-EGFR agents. These results are in line with studies on lung cancer cells with acquired resistance to erlotinib which could be overcome by the addition of cetuximab only partially [33]. Inversely, gefitinib and erlotinib retained anti-EGFR susceptibility in cetuximab-resistant head and neck and lung cancer cells [34]. Recently, in a clinical phase Ib study, patients with lung cancer and resistance to therapy with erlotinib and gefitinib showed a clinical benefit from sequential treatment, that is, after progression under therapy with the anti-EGFR TKI afatinib, they profited from the combined therapy with afatinib and cetuximab [35]. Other studies have shown that long-term treatment of ovarian cancer cells with the anti-Her2-antibody trastuzumab potentiated the responsiveness to gefitinib and cetuximab significantly, although the cells were functionally resistant to trastuzumab [36]. In our study, the abrupt withdrawal of TKI treatment revealed a rebound effect with overwhelming cell proliferation, which could be attenuated by cetuximab only temporarily. This corresponds to other data showing that lung and colorectal cancer cells with resistance to EGFR-related TKIs displayed a rebound effect in cell proliferation after the treatment with anti-MET TKIs was interrupted. Consecutively, specific anti-MET antibodies could weaken this rebound effect [37]. Furthermore, in the present study we showed that anti-EGFR TKI sensitization resulted in increased resistance to the cell death-inducing signal FasLigand. This finding could explain the development of functional anti-EGFR TKI resistance at least partially. Regarding the literature, different mechanisms of anti-EGFR TKI resistance have been extensively studied in lung cancer. Besides secondary mutations of the EGFR receptor [21] and MET amplification [37], activation of the Fas-/NFkB-pathway by anti-EGFR TKIs may also cause resistance to anti-EGFR agents [38]. In our studies, the expression of EGFR remained unaffected by anti-EGFR TKI treatment. However, selected published data suggest that anti-EGFR TKIs may reduce the degree of internalization of the EGF receptor [39] which may lead to altered receptor expression. In summary, our data indicate that EGFR-positive ovarian cancer cells show intrinsic or acquired resistance to anti-EGFR TKIs which cannot be abrogated by the anti-EGFR antibody cetuximab sufficiently. 

NK cells are major effectors of the innate immune system and display cytolytic activity without prior sensitization. Thus, NK cells may be a suitable partner in multimodal oncologic therapies. However, only limited data are available regarding the implication of the long-term application of anti-EGFR biologicals on the structure and function of NK cells as well as their interaction with tumor cells. In the present study, we showed that anti-EGFR sensitization of ovarian cancer cells enhanced antibody-dependent cellular cytotoxicity (ADCC) of co-incubated NK cells in the presence of cetuximab, which was correlated with increased tumor cell lysis. In contrast, natural cytotoxicity and tumor cell lysis in the absence of cetuximab remained unaffected. Our data are in agreement with in vitro and in vivo studies with non-small-cell lung cancer (NSCLC) cells which were sensitized by erlotinib for 24 h. Compared to unsensitized controls, sensitized cells in this study showed enhanced ADCC after adding cetuximab [40]. However, in our studies, the short-term application of anti-EGFR TKIs for several hours did not influence NK cell-mediated cytotoxicity [24]. There are other studies on colon and lung cancer cells which were sensitized with anti-EGFR TKIs for 24 h and 30 h, respectively. In these cells, sensitization even caused enhanced natural NK cell-mediated tumor cell lysis in the absence of a specific antibody [41,42]. Furthermore, in the present study we examined potential underlying mechanisms for enhanced NK cytotoxicity. We showed that the expression of MHC I molecules on the surface of the ovarian cancer cells was downregulated due to anti-EGFR sensitization. This finding is of particular importance since NK cells are mainly activated by cells losing their MHC I molecules on the cell surface. However, unexpectedly, natural cytotoxicity remained unchanged despite MHC I downregulation. Apparently, the presence of cetuximab seemed to be mandatory for enhanced NK cytotoxicity towards sensitized ovarian cancer cells. Regarding the literature, the modulation of MHC I receptors by anti-EGFR TKIs is supported by other data on non-small-cell lung cancer cells. However, in this study an increase of the MHC I expression on treated cells was reported [43]. 

In conclusion, based on our data on ovarian cancer, we can argue that anti-EGFR TKIs may indirectly enhance NK cytotoxicity via structural changes on ovarian cancer cells. However, in this context it would be expected that the expression of stress-induced NKG2D ligands such as MICA/B and ULBPs is also enhanced due to anti-EGFR sensitization. However, according to our data, anti EGFR sensitization resulted in a significant downregulation of MICA and B on both tested cell lines. Of note, similar data have been reported in the literature where erlotinib induced downregulation of MICA in lung cancer cells and still enhanced sensitivity to killer cells [44]. Supporting our hypothesis of indirect NK cell interference, we showed that the NK-activating cytokine IL18 is enhanced in the supernatants of sensitized ovarian cancer cells. However, we had no evidence that NK cells were directly affected by anti-EGFR TKIs. The activation receptor CD69 as well as NKG2D and Nkp44 remained unchanged by anti-EGFR TKIs. Therefore, anti-EGFR TKIs may act differentially compared to BCR/ABL multikinase inhibitors like dasatinib that is able to inhibit NK cytotoxicity efficiently by direct impairment of the NK cells’ signaling pathway [45]. 

Of note, our study demonstrated that NK cell function was differentially modulated. In contrast to cytolytic capacity, the production and release of IFNγ out of NK cells were suppressed in the presence of anti-EGFR TKI-sensitized ovarian cancer cells. Interestingly, the inhibition of cytokine release was even more distinctive in the absence of cetuximab. The presence of cetuximab seemed to attenuate the indirect interference of cytokine-producing NK cells by anti-EGFR TKIs. This finding is supported by Romee et al. showing that antibody-mediated NK cell stimulation via CD16 enhances the production of IFNγ [46]. Remarkably, the inhibitory potential of anti-EGFR TKIs on a specific NK cell function has not yet been published. However, some multityrosine kinase inhibitors like nilotinib are able to directly impair cytokine-producing NK cells by inducing cell death within the preferentially cytokine-secreting CD56(bright)CD16(-) NK cell subset [45]. 

Certainly, it is a limitation of our study that we did not test our therapeutic approaches using in vivo models. Such models offer the possibility to clarify the feasibility and the clinical relevance of the particular immunotherapeutic concept in vivo and to register potential adverse effects from the used components. In addition, patient-derived xenograft models could be used to assess the therapeutic effects of NK cells and targeted substances on human samples in an in vivo environment [47,48]. In the context of our study, it would be important to choose models that have the capacity to clearly distinguish human from murine NK activity. NK-deficient models, such as the common gamma chain (Il2rg) knockout mouse lacking NK cells and different cytokines or the double knockout mouse RAG2-Il2rg with the additional deficiency of T cells and B cells, would be suitable for such a prospective experimental in vivo design. 

In summary, our in vitro data show that EGFR-related TKIs and cetuximab are able to enhance antitumoral activity of NK cells in ovarian cancer cells, even if the tumor cells themselves are functionally resistant to the anti-EGFR agents. Thus, a combined immunotherapeutic approach in ovarian cancer consisting of targeted substances and immunocompetent NK cells may contribute to overcoming both functional resistance of tumor cells and their intrinsic resistance to NK cytotoxicity. This could also open new possibilities for chemo-immuno combination therapies in vivo which combine TKIs and NK activation strategies. According to our data, a sequential treatment with TKIs for several weeks followed by NK activation could be envisioned. However, at the same time, we also observed the reduced production of cytokines when NK cells were exposed to anti-EGFR TKI-sensitized ovarian cancer cells. Thus, future in vitro and in vivo studies are needed to fully elucidate the molecular mechanisms underlying the complex effects of anti-EGFR TKIs on the tumor–NK cell interaction. 

## 4. Materials and Methods 

### 4.1. Cell Lines and Cell Culture

The human ovarian cancer cell lines IGROV-1 and SKOV-3 were kindly provided by the Department of Obstetrics and Gynecology, University of Bonn, Germany. Both lines were cultured in RPMI-1640 supplemented with 10% FCS, 100 units/mL penicillin, and 100 µg/mL streptomycin. The tumor cells were cultivated in plastic culture flasks (Greiner, Solingen, Germany) at 37 °C and 5% CO_2_ and continuously passaged by treatment with StemPro^®^ Accutase^®^ (Invitrogen, Karlsruhe, Germany) for 5 min at 37 °C.

### 4.2. Sensitization of Ovarian Cancer Cells with Anti-EGFR TKIs

For anti-EGFR sensitization, the IGORV-1 and SKOV-3 tumor cells were cultured in RPMI-1640 supplemented with 10% FCS, 100 units/mL penicillin, and 100 µg/mL streptomycin. The anti-EGFR tyrosine kinase inhibitors (TKIs) erlotinib, gefitinib, and vandetanib (all from Selleck Chemicals, Houston, TX, USA) were reconstituted in DMSO (dimethyl sulfoxide, Carl-Roth-GmbH, Karlsruhe, Germany) before use. They were added to the culture in RPMI-1640 in a concentration of 1 µM which was titrated previously [24]. Cells were passaged and cultured continuously under anti-EGFR TKI treatment for 7 days and 6 weeks, respectively. 

### 4.3. MTT Proliferation Assay

The MTT colorimetric assay was performed to assess tumor cell viability under anti-EGFR-targeted therapy [49]. MTT (3-(4,5-dimethylthiazol-2-yl)-2,5-diphenyltetrazolium bromide) provides a measure of mitochondrial dehydrogenase activity and this metabolic activity correlates with cell number in culture. Anti-EGFR TKI-sensitized IGROV-1 and SKOV-3 ovarian cancer cells were seeded in 96-well plates (10,000 cells/well) (Greiner Bio-One, Frickenhausen, Germany) and treated with or without the monoclonal anti-EGFR antibody cetuximab (Erbitux^®^, ImClone Systems, Bristol-Myers Squibb, New York, NY, USA and Merck KGaA, Darmstadt, Germany) in a concentration of 1 µg/mL according to previously published data [24]. Thus, in the first experiments, TKI treatment was continued during the assay; in the following series of experiments, cell viability of sensitized ovarian cancer cells was assessed 72 h after cessation of TKI treatment. As the anti-EGFR TKIs erlotinib, gefitinib, and vandetanib had been reconstituted in DMSO before use, DMSO solvent alone was used as control. Unsensitized tumor cells treated with anti-EGFR TKI (1 µM) and cetuximab (1 µg/mL) served as controls. After the exposure time of 72 h at 37 °C and 5% CO_2_, the cell viability was assessed using the MTT assay. MTT (Sigma-Aldrich, Taufkirchen, Germany) was added for three to five hours. Cells were lysed by DMSO (dimethyl sulfoxide, Carl-Roth-GmbH, Karlsruhe, Germany) and analyzed photometrically. Data were collected as means of four to six replicates each. The percentage of cell viability was calculated using the following formula: 100/optical density of untreated cells × optical density of treated cells. 

### 4.4. Isolation of NK Cells of Healthy Donors

An amount of 100 to 150 mL peripheral blood per donor was obtained in citrate monovettes (Sarstedt AG & Co., Nümbrecht, Germany) and diluted (1:1, vol/vol) with phosphate-buffered saline (PBS). The mononuclear cell (MNC) fraction was collected and purified after separation by density centrifugation (Biocoll Separating Solution, Biochrom AG, Berlin, Germany) at 25 °C, 300× *g* for 30 min. Cells were counted by CasyCounter (Innovatis-Roche, Bielefeld, Germany). NK cells were isolated using the magnetic cell separator NK isolation kit II (Miltenyi Biotec, Mönchengladbach, Germany) according to the manufacturer´s protocol and used for the experiments immediately after isolation. Purity of cell subsets was routinely tested and ranged from 90% to 97%.

### 4.5. ELISA for Human IL18, IL12, and TNFα

We sensitized IGROV-1 and SKOV-3 cells with anti-EGFR TKIs as mentioned above and measured IL18, IL12, and TNFα in the supernatant of sensitized cells and unsensitized controls. IL18 was detected by an IL18-ELISA kit (Benders MedSystems, Vienna, Austria), and IL12 and TNFα using an R&D Systems kit (Wiesbaden, Germany). Both kits were used according to the manufacturer’s protocol.

### 4.6. Antibodies Used for Flow Cytometric Analysis (FACS)

Unsensitized and sensitized IGROV-1 and SKOV-3 tumor cells were characterized for the expression of EGFR, MHC class I polypeptide-related sequence A, B (MICA, MICB), MHC class I, UL16 binding protein 1 (ULPB 1), UL16 binding protein 3 (ULBP 3), UL16 binding proteins 2, 5 and 6 (ULBP 2, 5, und 6). Intracellular FACS staining was performed on NK cells for IFNγ, and extracellular staining on NK cells for CD69, Nkp44, and NKG2D. For FACS analysis, the following antibodies were used: anti-EGFR-PE (clone EGFR.1) and isotype mIgG2b-PE (both from BD Bioscience, Heidelberg, Germany), anti-MICA (clone AMO1) and anti-MICB (clone BAMO2) (both from Immatics, Tübingen, Germany), secondary antibody goat-anti-mouse PE (Agilent Technologies, Hamburg, Germany). Anti-HLA-class I-RPE (clone W6/32) and isotype mIgG2a-RPE (both from Agilent Technologies, Hamburg, Germany). Anti-ULBP-1-PE (clone 170818), anti-ULBP-3-PE (clone 166510), and anti-ULBP-2-5-6-APC (clone 165903) (all from Bio-Techne, Wiesbaden, Germany). Corresponding isotypes mIgG2a-PE and mIgG2a-APC were from BD Bioscience. Anti-CD69-FITC and isotype mIgG2a-FITC (both from Dianova, Hamburg, Germany), anti-NKG2D-PE (from R&D Systems) and isotype mIgG2a-PE (from BD Bioscience), anti-NKp44-PE (Beckman Coulter, Krefeld, Germany) and isotype mIgG2a-PE (BD Bioscience) were also used. For the intracellular staining of IFNγ, anti-IFNγ-APC (clone B27) from BD Bioscience was used. For the NK cell degranulation assay, anti-CD107a-FITC (clone H4A3) and isotype mIgG1-FITC (both from BD Bioscience) were used. Cells were analyzed on a FACS Canto II using Diva software 6.0 (Becton Dickinson, Heidelberg, Germany). 

### 4.7. Extracellular and Intracellular FACS Staining

For extracellular FACS staining 100,000 to 200,000 tumor cells were fixed and permeabilized using BD Cytofix/Cytoperm™ Plus according to the manufacturer’s protocol after staining with Fixable Viability Dye eFluor™ 780 (Invitrogen) to differentiate between vital and dead cells. Specific staining was performed in Perm/Wash buffer for 30 min at 4 °C. Isotype antibodies were used as controls. After staining, the cells were immediately analyzed on a FACS Canto II. 

### 4.8. CD107a Degranulation Assay

Since the lysosomal-associated membrane protein-1 (LAMP-1 or CD107a) in NK cells is expressed during degranulation and correlates with NK cell-mediated tumor cell lysis [50], the expression of CD107a on NK cells was used to evaluate natural and antibody-mediated NK cell cytotoxicity. TKI-sensitized and unsensitized IGROV-1 and SKOV-3 cells were co-incubated with purified unstimulated NK cells (1:1 cell ratio) on a flat-bottom 96-well microtiter plate (Greiner Bio-One). Cetuximab was directly added at 1 µg/mL in ADCC experiments. NK cells were labelled with anti-CD107a-FITC or isotype mIgG1-FITC 1:20, after one hour incubation at 37 °C in 5% CO_2_, GolgiStop monensin (BD GolgiStop, BD Bioscience) was added at a dilution of 1:600. After a further 5 h incubation at 37 °C in 5% CO_2_, cells were resuspended in 200 µL PBS with azide and immediately analyzed by flow cytometry.

### 4.9. 7-AAD/Annexin Staining

To assess tumor cell lysis, the Annexin-V Apoptosis Detection Kit (BD Bioscience) was applied and followed by FACS staining according to the manufacturer’s protocol. Adherent as well as suspension cells were harvested with StemPro^®^ Accutase^®^ (Invitrogen). After a washing step, all cells were resuspended in Annexin Binding Buffer containing Annexin-V-PE and 7-AAD. Cells were incubated at room temperature for 15min, washed, resuspended, and analyzed immediately. Sensitivity of sensitized tumor cells to FasLigand and TRAIL was assessed by using the Annexin-V Apoptosis Detection Kit after sensitizing the IGROV-1 cells with erlotinib as mentioned above for 7 days and exposure to recombinant FasLigand (Alexis, Grünberg, Germany) and TRAIL (Merck, Darmstadt, Germany) in increasing concentrations (0, 10, 30, and 100 ng/mL) for 24 h.

### 4.10. ELISpot for IFNγ Detection

For the sensitive detection of IFNγ-secreting NK cells, the ELISpot (Enzyme-Linked Immunospot Assay) technique was applied. On the day before the assay, the membranes of the 96-well plates (MultiScreen Filter plates, Merck, Millipore, Darmstadt, Germany) were first activated with ethanol and then coated with the anti-IFNγ capture antibody (IFNγ clone 1-D1K, Mabtech, Nacka Strand, Sweden). After overnight incubation at 4 °C, the plates were blocked by the addition of 180 µL of RPMI-1640 media supplemented with 10% FCS, 100 units/mL penicillin, and 100 µg/mL streptomycin for 2 h at 37 °C. Plates were washed with PBS and loaded with NK cells co-incubated with sensitized tumor cells in the presence of cetuximab. Unsensitized cells, negative controls (tumor cells only), and positive controls (IL15-/IL18-stimulated NK cells) were included. After 20 h incubation at 37 °C, the plates were washed with ELISpot washing buffer (0.05% Tween-20 in PBS). Biotinylated anti-IFNγ detection antibody (IFNγ-Biotin clone 7-B6-1, Mabtech) in 1 mg/mL PBS + 5% BSA was added and plates were incubated for 2 h at 37 °C, then subsequently washed and incubated with ExtraAvidin alkalinephosphatase (Sigma-Aldrich) (1:1000 in PBS + 0.5% BSA) for 1 h at room temperature. Afterward, substrate BCIP/NBT (Sigma-Aldrich) was added and cytokine spots developed and were counted using an ELISpot reader (AID ELISpot reader system, AID Diagnostica GmbH, Straßberg, Germany). IFNγ Spot numbers were determined from triplicate wells/cell population by subtracting the mean number of ELISPOTs in the unstimulated wells.

### 4.11. Statistical Analysis

Results are expressed as means and standard deviation (SD) of several independent experiments. For statistical evaluation, the unpaired *t*-test was performed and statistical significance was assumed at a level of *p* ≤ 0.05. The statistical calculations and illustrations were performed using SigmaPlot for Windows Version 11 (Systat Software GmbH, Erkrath, Germany) and Adobe Illustrator (Adobe Systems Software, Dublin, Ireland).

## Figures and Tables

**Figure 1 ijms-20-04693-f001:**
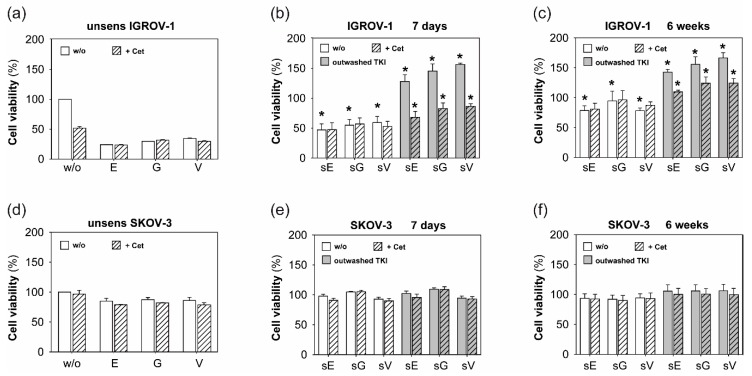
Effects of sensitization with anti-epidermal growth factor receptor (EGFR) tyrosine kinase inhibitors (TKIs) on ovarian cancer cell viability and antiproliferative activity of cetuximab (Cet). Cell viability (%) of (**a**) IGROV-1 and (**d**) SKOV-3 after short-term treatment (72 h) with erlotinib (E), gefitinib (G), and vandetanib (V) is shown in white columns. Striped columns indicate addition of cetuximab. Cell viability was assessed by performing MTT assay. (**b**,**c**,**e**,**f**): Cell viability of (**b**+**c**) IGROV-1 and (**e**+**f**) SKOV-3 sensitized with erlotinib (sE), gefitinib (sG), and vandetanib (sV) (1 µM) for 7 days and 6 weeks, respectively. White columns illustrate cell viability under continuous anti-EGFR treatment, grey columns document cell viability 72 h after discontinuation of TKI treatment. Striped columns show addition of cetuximab. Bars show means +/- standard deviation (SD) of four to six independent experiments. For illustration purposes, the cell viability of treated and sensitized cells is shown in relation to untreated and unsensitized tumor cells (w/o) in (**a**) and (**d**). Unpaired *t*-test was applied for statistical analysis, statistical significance (*p* < 0.05) is indicated (*).

**Figure 2 ijms-20-04693-f002:**
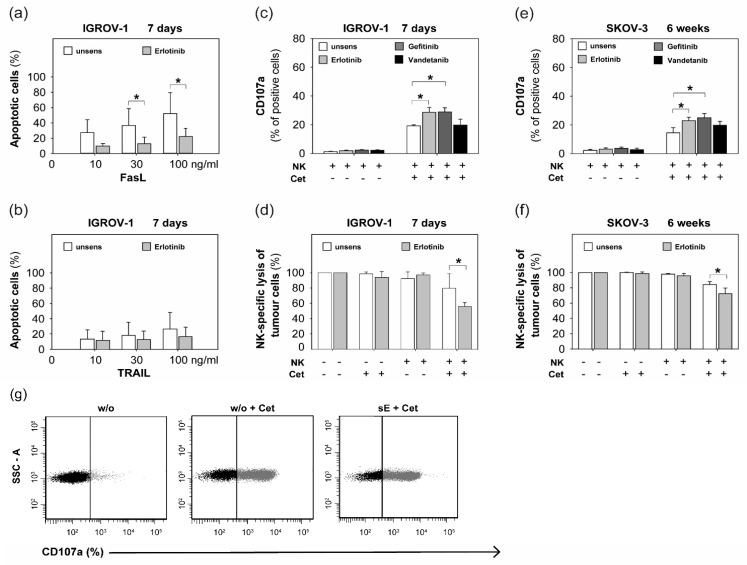
Sensitivity of anti-EGFR TKI sensitized ovarian cancer cells to FasLigand, TRAIL, NK-mediated cytotoxic degranulation, and NK cell-related lysis. Percentage of apoptotic cells (%) of erlotinib-sensitized IGROV-1 cells (7 days) and unsensitized controls after exposure to **(a)** FasLigand (FasL) and (**b**) TRAIL for 24 h in increasing concentrations up to 100 ng/mL. Analysis per FACS after performing Annexin-V Apoptosis Detection Kit. (**c**)–(**f**): Anti-EGFR-TKI sensitization of IGROV-1 for 7 days and SKOV-3 for 6 weeks. Co-incubation (1:1 cell ratio) with NK cells isolated from healthy donors with or without cetuximab (1 µg/mL). (**c**) + (**e**): NK cell-mediated cytotoxic degranulation: CD107a-positive NK cells (%) after performing CD107a degranulation assay and analyzing per FACS. (**d**) + (**f**): NK-specific tumor cell lysis. Tumor cell viability (%) as difference between vital and apoptotic cells in relation to unsensitized controls (= 100%) after performing Annexin-V Apoptosis Detection Kit and analyzing in the flow cytometer. Means +/- SD of at least three independent experiments are shown. Statistical analysis was performed by unpaired *t*-test, statistical significance (*p* < 0.05) is indicated (*). (**g**) Plots of the percentage of CD107a-positive NK cells in the presence of unsensitized IGROV-1 cells without (w/o) or with cetuximab (w/o + Cet) and in the presence of erlotinib-sensitized IGROV-1 cells (7 days) and cetuximab (sE + Cet). A representative experiment is shown.

**Figure 3 ijms-20-04693-f003:**
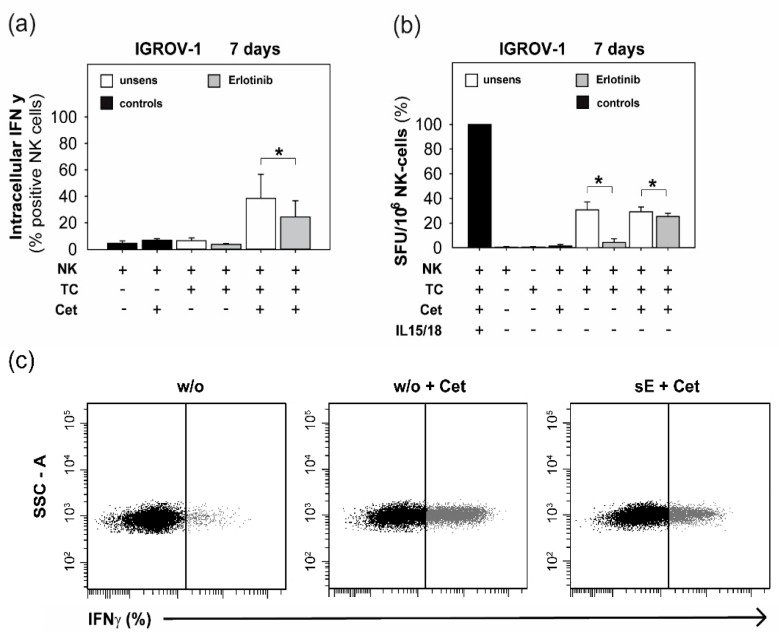
IFNγ production and release of NK cells in the presence of erlotinib-sensitized IGROV-1 cells. NK cells of healthy donors were co-incubated in a 1:1 cell ratio with erlotinib-sensitized IGROV-1 cells (7 days) with or without cetuximab (1 µg/mL). (**a**) Intracellularly IFNγ-positive NK cells (%) after APC-coupled anti-IFNγ staining or isotype control and analysis per FACS. (**b**) IFNγ release out of NK cells was quantified via ELISpot. IL15/IL18-stimulated NK cells served as positive controls. Means +/- SD of at least three independent experiments are shown. Statistical analysis was performed by unpaired *t*-test, statistical significance (*p* < 0.05) is indicated (*). (**c**) Plots of the percentage (%) of IFNγ-positive NK cells in the presence of unsensitized IGROV-1 cells without (w/o) or with cetuximab (w/o + Cet) and in the presence of erlotinib-sensitized IGROV-1 cells (7 days) and cetuximab (sE + Cet). A representative experiment is shown.

**Figure 4 ijms-20-04693-f004:**
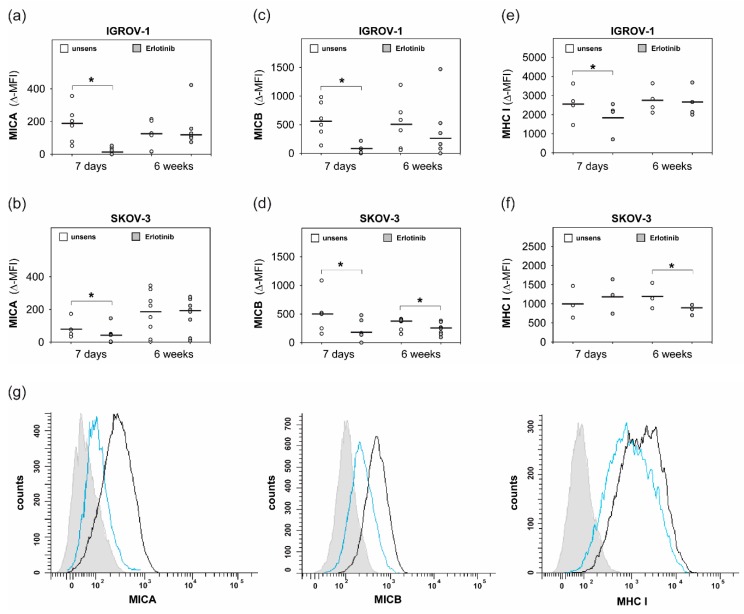
Expression of MHC class I polypeptide-related sequence A, B (MICA, MICB) and MHC I on erlotinib-sensitized ovarian cancer cells. IGROV-1 and SKOV-3 were sensitized with erlotinib (1 µM) for 7 days and 6 weeks. Treated cells and unsensitized controls were stained with anti-MICA/anti-MICB and secondary antibody IgG1-PE and anti-HLA-class I-RPE and isotype controls, respectively. Analysis per FACS. ∆-MFI values show the difference of MFI between specific antibody and isotype control. (**a**) + (**b**): Expression of MICA. (**c**) + (**d**): Expression of MICB. (**e**) + (**f**): Expression of MHC I. Means +/- SD of at least three independent experiments are shown. Statistical analysis was performed by unpaired *t*-test, statistical significance (*p* < 0.05) is indicated (*). (**g**) Histogram: erlotinib-sensitized IGROV-1 (7 days) (blue line) or unsensitized controls (black line) were stained with anti-MICA/anti-MICB and secondary IgG1-PE or anti-HLA-class I-RPE, respectively, or isotype control (filled histogram). MFI of a representative experiment is shown.

**Figure 5 ijms-20-04693-f005:**
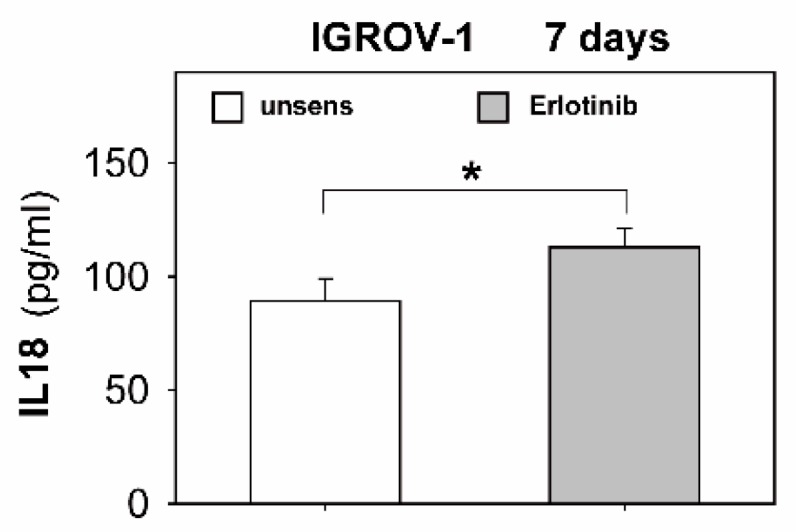
Secretion of IL18 of erlotinib-sensitized IGROV-1 cells. IGROV-1 cells were sensitized with erlotinib (1 µM) for 7 days. Secretion of IL18 of sensitized cells and untreated controls was quantified in the supernatants by ELISA. Bars show IL18 in pg/mL per 10^6^ tumor cells. Means +/- SD of three independent experiments are shown. Statistical analysis was performed by unpaired *t*-test, statistical significance (*p* < 0.05) is indicated (*).

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
