# Peer review of "EGFR-Specific Tyrosine Kinase Inhibitor Modifies NK Cell-Mediated Antitumoral Activity against Ovarian Cancer Cells"

_ijms, 2019, doi:10.3390/ijms20194693_

Round 1
Reviewer 1 Report
This excellent work explores the effect of tyrosine kinase Inhibitors (TKIs) and Cetuximab on NK cell mediated Tumor cell Lysis of ovarian cancer cells. Long-term exposure to the TKIs Erlotinib, Gefitinib and Vandetanib led to a reduced sensitivity of the cancer cells. However, these long-term treated cancer cells showed an increased sensitivity against NK-cells in combination with Cetuximab. This effect can be explained by an increased ADCC under the combined exposure with NK cells and Cetuximab, Meanwhile, the cytokine production was reduced under this treatment combination.
All results are sound and the conclusions drawn are convincing. The linguistic style is very good which makes the manuscript easy to read and understand. The role of NK cells in cancer is recently getting more and more awareness. Therefore, this work will be of high interest to many readers. Summarizing, I strongly support publication in IJMS without further changes needed.
Author Response
We really thank the reviewer for his positive assessment.
Reviewer 2 Report
The authors demonstrated that EGFR-specific TKI can modify antitumoural effect of NK cells for two human ovarian cancer cells such as IGROV1 and SKOV3 in vitro analyses with human ovarian cancer cell lines. However there are some problems to be solved to accept this article in IJMS.
# The authors should show antitumor effects (survival analysis and TIL immunohistochemical analyses) of EGFR-TKI with NK cells in syngeneic mouse ovarian cancer model such as ID8/BL6 mice or HM1/B6C3F1mice, in vivo.
And humanized mouse ovarian cancer PDX model treated with patient derived NKcells and EGFR-TKI treatment.
The authors should show why EGFR-TKI collaborated with NK cells activity, please discuss it more in detail.
Author Response
1.) The authors should show antitumor effects (survival analysis and TIL immunohistochemical analyses) of EGFR-TKI with NK cells in syngeneic mouse ovarian cancer model such as ID8/BL6 mice or HM1/B6C3F1mice, in vivo.
2.) And humanized mouse ovarian cancer PDX model treated with patient derived NK cells and EGFR-TKI treatment.
1.) and 2.): Our study is an in vitro study, which looks at the consequences of sensitization of ovarian cancer cells by TKI on the subsequent NK cell response and activity. While humanized mouse models or PDX models are well suited for some research questions, these studies would extend the current study far beyond its current design and scope. According to German law the extension to the murine model level would require animal ethics application, review and approval. This would take several months in the state of NRW, Germany, and will make a revision in a reasonable time impossible. For this reason, it is impossible for us to address this main aspect.
3.) The authors should show why EGFR-TKI collaborated with NK cells activity, please discuss it more in detail.
Please find our changes in the discussion. We explained the collaboration of NK cells and anti-EGFR-TKI more in detail.
Reviewer 3 Report
The authors of the manuscript "EGFR-specific tyrosine kinase inhibitor modifies NK cell mediated antitumoural activity against ovarian cancer cells" evaluated the Cetuximab-treatment effects of two cell lines previously exposed to long-term treatment with TKI. Actually some data and some experiments appear not so well connected to each others and to the main goal of the study.
The current version of manuscript is not acceptable for publication.
Major points:
1. First of all, the rationale of the study is not very clear. Why should long term exposure to TKI enhance responsiveness to Cetuximab? It does not increase EGFR expression as reported on page 4 line 135, what about EGFR pathway activation?
2. Second, why did the authors call "sensitization" the long-term treatment with TKI if it actually reduce the responsiveness and induce resistance?
3. Figure 1a-f misses Statistical analysis. Figure 1b-c and e-f: cell viability is expressed in % and so it is a relative measure, but It is % respect to what? There is no control in these panels.
4. Figure 2a and 2b: it is more correct to describe % of apoptosis than % of cell viability. Please re-analyze the data and show them as % of AnnexV+/PI- and AnnexV+/PI+ cells. Figure2e: How can the authors explain the choise of the "sensitization" time of SKOV-3 for degranulation assay given that in Figure 1 there were no differences between 7 days (e) and 6 weeks (f)? In Figure 2d and 2f is described specific lysis by NK or cell viability? It is not specific and correct to call everything "cell ciability". why do you chose unpaired t-test for statistical analysis instead of paired one?
5. Figure 3a and b. Define TZ. what is it? why do you use unpaired t-test?
6. Figure 4a-f : which is the meaning of Δ-median (median of MFI?)
7. In all figures regarding flow cytometry data lack flow cytometry plot or histogram of one representative experiment.
8. The discussion in the current form is intricate, where possible make it more easily readable and fluid.
Minor points:
1. pag 6 line 225: why do you decide not to show results of cytokine secretion, especially of IL-18, given they are significant?
2. Review the figure legend and make them more clear
3. Review English and typos.
Author Response
The authors of the manuscript "EGFR-specific tyrosine kinase inhibitor modifies NK cell mediated antitumoural activity against ovarian cancer cells" evaluated the Cetuximab-treatment effects of two cell lines previously exposed to long-term treatment with TKI. Actually, some data and some experiments appear not so well connected to each other’s and to the main goal of the study.
Major points:
1.) (a) First of all, the rationale of the study is not very clear. (b) Why should long term exposure to TKI enhance responsiveness to Cetuximab? (c) It does not increase EGFR expression as reported on page 4 line 135, what about EGFR pathway activation?
(a) The rationale of the study consists of two parts. First, we examined the consequences of long-time-exposure of anti-EGFR-TKI to EGFR-positive ovarian cancer cells which are either sensitive or intrinsically resistant to the anti-EGFR-antibody Cetuximab. Second, we investigated the NK cell activity to TKI-exposed tumour cells.
Our study showed that anti-EGFR-sensitive ovarian cancer cells develop resistance to anti-EGFR-TKI which cannot be compensated by the antibody Cetuximab. In intrinsically anti-EGFR-resistant ovarian cancer cells the pre-treatment with anti-EGFR-TKI was not able to restore sensitivity to Cetuximab. Of note, co-incubated NK cells displayed significantly increased cytotoxic activity towards pre-treated tumour cells in presence of Cetuximab while cytokine secretion was reduced.
As therapeutic TKI are usually administered as oral drugs for several weeks and months, our in vitro study provides elementary findings for the clinical use of anti-EGFR-TKI. Especially, the potential impact of anti-EGFR-TKI on the interaction between tumour cells and NK cells should be the subject for further examination.
(b) Indeed, there are studies in other cancer entities showing that exposure to specific TKI enhance responsiveness to monoclonal antibody. So Gefitinib and Erlotinib retained anti-EGFR-susceptibility to Cetuximab-resistant head and neck as well as lung cancer cells [36]. Furthermore, patients with lung cancer showed benefit from sequential treatment with the anti-EGFR-TKI Afatinib in combination with Cetuximab beyond progression on Afatinib in a clinical phase Ib study [37]. In ovarian cancer cells with resistance to anti-Her2-antibody Trastuzumab long-term treatment with Trastuzumab potentiated the responsiveness to Gefitinib and Cetuximab significantly [38].
(c) In our study the treatment of ovarian cancer cells with anti-EGFR-TKI did not change the EGFR-expression. But there are other published data which show that anti-EGFR-TKI reduce the degree of internalization of the EGFR with consecutive altered receptor expression [41]. As in our cells the EGFR-expression remained unchanged by TKI we did not further investigated changes regarding the EGFR-pathway.
2.) Second, why did the authors call "sensitization" the long-term treatment with TKI if it actually reduce the responsiveness and induce resistance?
The term “sensitization” aims to illustrate the enhanced cytotoxic NK activity to TKI-treated tumour cells. The developed resistance of tumour cells to TKI is partially compensated by enhanced NK mediated tumour cell lysis.
The focus of our manuscript is on the altered NK activity to TKI-pre-treated ovarian cancer cells.
3.) (a) Figure 1a-f misses Statistical analysis. (b) Figure 1b-c and e-f: cell viability is expressed in % and so it is a relative measure, but It is % respect to what? There is no control in these panels.
(a) We added statistical analysis and indicated statistical significance (*) in panel (b) and (c). We completed and reviewed Figure legend 1.
(b) The cell viability of the untreated and unsensitized cells (in a and d) is set to 100% and serves as a reference for the following sensitized cells (b+c and e+f). We added this aspect in the figure legend.
4.) (a) Figure 2a and 2b: it is more correct to describe % of apoptosis than % of cell viability. Please re-analyze the data and show them as % of AnnexV+/PI- and AnnexV+/PI+ cells. (b) Figure2e: How can the authors explain the choice of the "sensitization" time of SKOV-3 for degranulation assay given that in Figure 1 there were no differences between 7 days (e) and 6 weeks (f)? (c) In Figure 2d and 2f is described specific lysis by NK or cell viability? It is not specific and correct to call everything "cell viability". why do you choose unpaired t-test for statistical analysis instead of paired one?
(a) We agree with the reviewer that in Fig. 2a and 2b percentage of apoptotic cells is examined and we thank for the suggestion to show the data the other way around. However, we consciously decided to show this data also in kind of “cell viability” in order to create direct comparison between the different readouts in the Figures 1 and 2.
(b) The “sensitization” time basically was chosen for evaluating the differences between “days” and “weeks”. It might be assumed that the time of 6 weeks represents long-lasting drug influence sufficiently. As the proliferation of anti-EGFR-sensitive IGROV-cells is considerably impaired by TKI longer time of treatment (e.g. several months) would not have led to adequate quality of experiments.
According to the conditions for IGROV-1 we also chose the same “sensitization” time” for the anti-EGFR-resistant cell line SKOV-3 and for all read outs in order to assure comparability. Fig. 2 e+f impressively shows that NK activity might be significantly altered by TKI-treated SKOV-3-cells even if their proliferation remains unchanged (Fig.1).
(c) We thank the reviewer for this comment. But as explained above (a) we aimed to create direct comparison between different readouts in the figures.
We chose unpaired t-test because treated cells are independent from the untreated controls. They were seeded and cultured in parallel culture flasks under same conditions with and without TKI-treatment. We assume that the culture time for several days and weeks may lead to changes in cell properties in different directions.
5.) Figure 3a and b. (a) Define TZ. what is it? (b) why do you use unpaired t-test?
(a) We thank the reviewer for this note. Of course, we change it to “TC=Tumour cell”.
(b) see above under 4(c).
6.) Figure 4a-f: which is the meaning of Δ-median (median of MFI?)
Δ-median means MFI, “Δ" means the difference between fluorescence intensity of specific antibody and isotype control. We change the labelling to Δ-MFI and explain it in the figure legend to clarify this aspect.
7.) In all figures regarding flow cytometry data lack flow cytometry plot or histogram of one representative experiment.
We thank the reviewer for this suggestion. We added dot plots and histograms of one representative experiment each in all figures regarding flow cytometry which will illustrate the data additionally.
8.) The discussion in the current form is intricate, where possible make it more easily readable and fluid.
We revised and completed all parts of the discussion thoroughly and made it more easily readable where it was possible.
Minor points:
1.) pag 6 line 225: why do you decide not to show results of cytokine secretion, especially of IL-18, given they are significant?
We added Fig. 5 with results of IL18 under section 2.4.
2.) Review the figure legend and make them more clear
All figure legends were rewritten and completed.
3.) Review English and typos.
We thoroughly reviewed the manuscript and edited English language and style. All typos were corrected.
Round 2
Reviewer 2 Report
The revised version of this paper has not been improved There is no in vivo mouse tumor treatment model nor experiments of clinical samples.
Author Response
The revised version of this paper has not been improved There is no in vivo mouse tumor treatment model nor experiments of clinical samples
Due to German law with the requirement of ethics application for animal experiments or patients studies we are not able to address this critical aspect.
For this, we added a section about limitation of the study in the discussion.
Reviewer 3 Report
In my opinion it is more correct to indicate each experiment with the adequate name. If you evaluate apoptosis, you have to indicate the percentage of apoptotic cells. If you evaluate the Nk-mediated cell lysis, you have to write the % of specific lysis. I think that the reason which you mentioned about the possibility to compare different graphs in the panels it is not acceptable. A careful reader is able to understand the meaning of different graphs.
In fig. 1b and 1c please indicate in the results depicted with grey columns and grey striped columns are significant or not
In Fig.2g and 3c please insert % into the graphs
Author Response
1.) In my opinion it is more correct to indicate each experiment with the adequate name. If you evaluate apoptosis, you have to indicate the percentage of apoptotic cells. If you evaluate the Nk-mediated cell lysis, you have to write the % of specific lysis. I think that the reason which you mentioned about the possibility to compare different graphs in the panels it is not acceptable. A careful reader is able to understand the meaning of different graphs.
We followed the reviewer´s suggestions and show the percentage of apoptotic cells in Figure 2a and b. Furthermore, in Figure 2d and 2f we changed the labelling to “ NK-specific tumour cell lysis [% ] ”.
2.) In fig. 1b and 1c please indicate in the results depicted with grey columns and grey striped columns are significant or not
We completed the illustration of the significant results in Figure 1b and 1c
3.) In Fig.2g and 3c please insert % into the graphs
We inserted “ [%] “ into the graphs of Figure 2g and 3c.